# Intra-operative autologous blood donation for cardiovascular surgeries in Japan: A retrospective cohort study

**Takuya Okuno**[1], **Susumu Kunisawa**[1], **Kiyohide Fushimi**[2], **Yuichi Imanaka**[1]*

**1** Department of Healthcare Economics and Quality Management, Graduate School of Medicine, Kyoto University, Kyoto, Japan, **2** Health Policy and Informatics Section, Graduate School of Tokyo Medical and Dental University, Tokyo, Japan

* imanaka-y@umin.net

**Data Availability Statement:** The data generated and analyzed during this study cannot be shared publicly, due to Ethical Guidelines for Medical and Health Research Involving Human Subjects of the

## Abstract

Intra-operative autologous blood donation is a blood conservation technique with limited evidence. We evaluated the association between intra-operative autologous blood donation and decrease in peri-operative transfusion in cardiovascular surgery based on evidence from a Japanese administrative database. We extracted the data of patients who had undergone cardiovascular surgery from the Diagnosis Procedure Combination database in Japan (2016–2019). Based on the surgery type, we examined the association of intra-operative autologous blood donation with the transfusion rate and amount of blood used in cardiac and aortic surgeries using multilevel propensity score matching. We enrolled 32,433 and 4,267 patients who underwent cardiac and aortic surgeries and received 5.0% and 6.7% intra-operative autologous blood donation with mean volumes of 557.68 mL and 616.96 mL, respectively. The red blood cell transfusion rates of the control and intra-operative autologous blood donation groups were 60.6% and 38.4%, respectively, in the cardiac surgery cohort (p < .001) and 91.4%, and 83.8%, respectively, in the aortic surgery cohort (p = .037). The transfusion amounts for the control and intra-operative autologous blood donation groups were 5.9 and 3.5 units of red blood cells, respectively, for cardiac surgery patients (p < .001) and 11.9 and 7.9 units, respectively, for aortic surgery patients (p < .001). Intra-operative autologous blood donation could reduce the transfusion rate or amount of red blood cells and fresh frozen plasma for patients undergoing index cardiovascular surgery and could be an effective blood transfusion strategy in cardiovascular surgery for Japanese patients.

## Introduction

More than one-half of cardiovascular surgery (CVS) patients receive some type of blood transfusion [1]. In Japan, 16.7% of red blood cell (RBC) transfusions and 28.6% of fresh frozen plasma (FFP) transfusions are related to cardiovascular surgery [2]. Efforts should be made to decrease the amount of blood transfusion, considering adverse effects and limited resources.

Ministry of Health, Labour and Welfare, Japan (a provisional translation is available from https://www.mhlw.go.jp/file/06-Seisakujouhou-10600000-Daijinkanboukouseikagakuka/0000080278.pdf). However, other researchers may send data access requests to the Ethics Committee, Graduate School of Medicine, Kyoto University (ethcom@kuhp.kyoto-u.ac.jp).

**Funding:** Yuichi Imanaka received Health Sciences Research Grants from the Ministry of Health, Labour and Welfare of Japan (20AA2005) and a Grant-in-Aid for Scientific Research from the Japan Society (19H01075).

**Competing interests:** The authors have declared that no competing interests exist.

Allogenic RBCs are associated with increased infection risk, renal failure, respiratory failure, stroke, myocardial infarction, and mortality after CVS [3–5]. Moreover, because of the coronavirus disease 2019 (COVID-19) pandemic, blood supply shortage is becoming an important issue globally [6]. The World Health Organization also announced guidelines on protecting blood supply during the COVID-19 pandemic [7].

In CVS, which consumes large amounts of donated blood, efficient transfusion can ensure the proper use of the limited medical resources and improve patients' outcomes. Some conservation techniques, such as targeted transfusion guidelines, cell salvage, and retrograde autologous priming, are available for maintaining hemoglobin levels [8–10].

Another solution is intra-operative autologous blood donation (IAD), also known as acute normovolemic hemodilution [11]. Although the evidence level of IAD was IIB [12], which was not high, the Ministry of Health, Labour and Welfare of Japan approved the procedure in 2016, adding to the universal benefit scheme. Changing the blood conservation policy to allow IAD use can improve transfusion practices in CVS [13]. In retrospective studies [14, 15], patients with a large volume of IAD ($\geq$800 mL or $\geq$900 mL) received fewer peri-operative RBC transfusions than patients who received no or fewer IADs. However, high-volume of IAD can be administered only to patients who have sufficient circulation volumes, such as those with obesity or body surface area (BSA) of >2 m$^2$ [15]. Moreover, the patients' characteristics were not well balanced, even after propensity matching, based on the standard mean difference (SMD) of 0.1 [14]. In this study, we examined the association between IAD and decrease in peri-operative transfusion for index CVS in multiple hospitals using administrative data of Japanese individuals, who are generally physically small.

## Patients and methods

The study protocol was approved by the Ethics Committee of Kyoto University Graduate School, Kyoto, Japan (R0135). This study was conducted in accordance with the ethical guidelines for medical and health research involving human participants issued by the Japanese Government. The data were anonymized, and the requirement for informed consent was waived.

### Data sources

This was a retrospective cohort study based on the Diagnosis Procedure Combination (DPC) database, which is a national administrative Japanese database and comparable to diagnosis-related databases in the United States [16, 17]. The database covers >1500 hospitals and accounts for >80% of tertiary care emergency hospitals in Japan. The database does not include information on laboratory findings, cardiac function, or details about surgery; instead, it provides data from the discharge summary, such as weight, primary diagnosis, comorbidities (identified using the International Classification of Diseases 10th Revision [ICD-10] codes), surgery type, cardiopulmonary bypass (CPB) duration during CVS, drug or device prescriptions, and codes corresponding to the medical procedures performed. We accessed the database and got the information of eligible patients in July 2020.

### Study population

Patients aged 18–84 years, with a body weight of >40 kg who had undergone scheduled CVS using CPB between April 2016 and March 2019 were eligible for the study. The target CVSs were isolated coronary artery bypass grafting ([CABG] K5521, K5522), isolated valve repair (K554, K555), CABG + valve repair, isolated ascending aortic repair (K561), isolated arch repair (K562), and ascending + arch repair (K563). Overall, 38,981 patients underwent these surgeries. We excluded 157 patients because the CPB time was too short, indicating possible

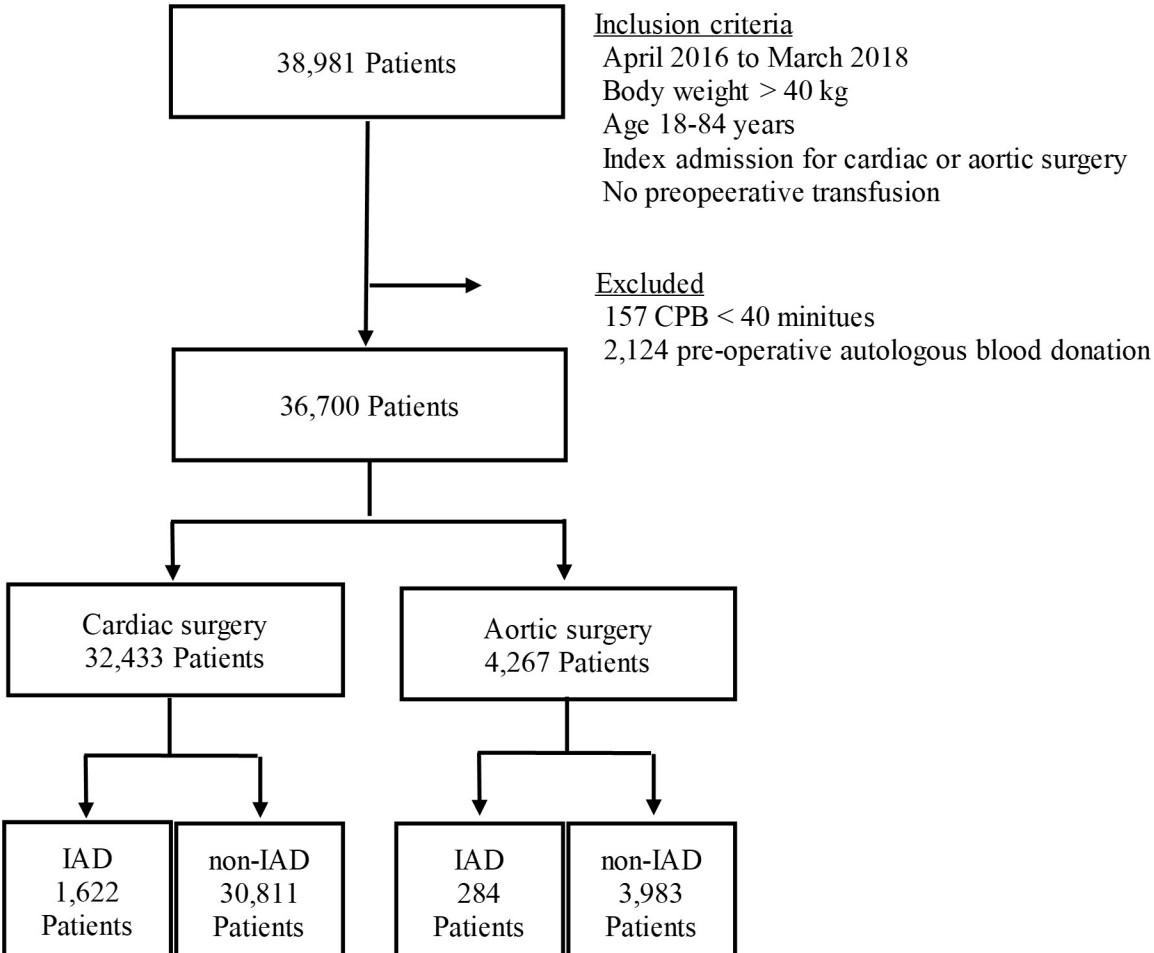

**Fig 1. Flowchart depicting the selection of patients.** CPB, cardiopulmonary bypass; IAD, intra-operative autologous blood donation.

intra-operative challenges (<40 min), based on the exclusion criteria of a previous study [14]. We also excluded 2,124 patients who received pre-operative blood donation. Finally, we analyzed 32,433 CVS and 4,267 patients who had undergone cardiac surgery and aortic surgery, respectively (Fig 1). The intervention group included patients who received IAD, and the control group included those who did not receive IAD. In this target population, no patient underwent multiple cardiovascular surgeries during their admission. For patients with a history of several index CVS admissions, only the first event was included.

The primary outcome was the allogenic blood transfusion rate (RBC, FFP, and platelets [PLT]) during hospitalization. The secondary outcome was the amount of allogenic transfusion units (i.e., RBC, FFP, and PLT) during admission. We also aggregated data for re-operation due to bleeding during admission; post-operative adverse events such as acute kidney injury, respiratory failure, infection (i.e., urinary, respiratory, and surgical site); and post-operative intensive care unit (ICU) stay.

## Variables

We extracted the patients' demographic characteristics, specific comorbidities (ischemic heart disease [ICD-10: I20-25], aortic stenosis [ICD-10: I062, I350, and I352], heart failure [ICD-10:

I110, I420, I421, and I50], renal failure [ICD-10: N03, N04, N17-N19, R34, I12, I13, or Z992], diabetes with insulin use, anemia [ICD-10: D50-59 and D60-64], coagulation disorder [ICD-10: D65-69, K703, K704, K72, and K743-746]), bridging anticoagulation, pre-operative iron use, CVS history, and the amount of pre-operative autologous blood donation. We also considered the interval of intra-operative CPB by the temperature, deep and moderate hypothermia (<32˚C), and mild hypothermia (≥32˚C). The CPB of all cardiac surgeries was used more often than moderate hypothermia, and the CPB of some aortic surgeries was performed under deep hypothermia. Perioperative tranexamic acid was used for all patients.

## Statistical analysis

We assigned patients with claims of additional fees for IAD in the IAD group and those without these claims in the control group. We performed propensity score matching for clustered data because the Diagnosis Procedure Combination data were considered to have a clustered structure of patients nested in each hospital. The covariates for estimating the propensity scores included sex, age, body weight, pre-operative heparin use, pre-operative iron use, procedure category, previous cardiac operation, and CPB time by temperature. The covariates of comorbidities were selected based on recommendations in previous literature reports [13–15]. Fixed-effects propensity scores were estimated by fitting the multivariable logistic regression models with the listed covariates and dummy variables of each hospital as the intercepts [18]. The dependent variables of the propensity score model were the indicator variables of the claims for additional fees of IAD. The fixed-effects propensity scores were used for fixed-effects matching. Patients in the IAD group were matched with those in the control group. For all matching, we performed one-to-one nearest-neighbor matching with replacement using a caliper of 0.2, as recommended by Austin [19]. The covariate balance was determined using the SMD. A covariate was considered to have a good balance when the SMD was <0.1. The primary and secondary outcomes were compared between groups using the paired $t$-test for continuous variables and the McNemar test for categorical variables. For all statistical tests, a two-sided p-value < .05 was considered significant. Statistical analysis was conducted using R software (version 3.5.3, R Foundation for Statistical Computing, Vienna, Austria). For the sensitivity analysis, we conducted the same analysis on CVS patients who received no or <800 mL IAD based on the previous reports [14, 15].

## Results

Between April 2016 and March 2019, 38,824 patients from the Diagnosis Procedure Combination database that had undergone CVS were analyzed. Comparing before and after fixed-effect propensity score matching of patients' demographics, comorbidities and surgical information of cardiac surgery is presented in Tables 1 and 2, and that of aortic surgery in Tables 3 and 4. Before matching, the mean age and mean body weight were larger in the IAD group. For the IAD group, the mean IAD dose was 557.68 mL and 616.96 mL for the cardiac surgery and aortic surgery, respectively. After matching, all variables were well balanced with SMD < 0.1. Particularly, the CPB time by the temperature, which was a significant factor for post-operative bleeding, transfusion, and other morbidities in CVS, was well balanced [19]. Before fixed-effect propensity score matching, 76.7% of cardiac surgery patients and 94.8% of aortic surgery patients received RBC transfusions during their hospitalization. After matching, the RBC transfusion rates of the control and IAD groups were 60.6% and 38.4%, respectively (p < .001), in the cardiac surgery cohort, and 91.4% and 83.8% (p = .037), respectively, in the aortic surgery cohort. The risk difference of the FFP transfusion rate was statistically lower in the patients who had IAD (cardiac surgery: risk difference = -18.7%, p < .001; aortic surgery: risk

**Table 1. Baseline characteristics and comorbidities of the cardiac surgery patients before and after multilevel propensity score matching.**

|  | Before matching | | | After matching | | |
|---|---|---|---|---|---|---|
|  | **Control** | **IAD** | **SMD** | **Control** | **IAD** | **SMD** |
| n | 30,811 | 1,622 |  | 1,233 | 1,233 |  |
| Sex (male), n (%) | 18,980 (61.6) | 1,196 (73.7) | 0.262 | 908 (73.6) | 914 (74.1) | 0.011 |
| Age (y), mean (SD) | 69.26 (11.07) | 65.07 (12.76) | 0.351 | 64.73 (13.59) | 65.28 (12.33) | 0.042 |
| Body weight (kg), mean (SD) | 59.49 (11.52) | 64.27 (12.66) | 0.395 | 63.31 (12.50) | 64.01 (11.96) | 0.057 |
| Ischemic heart disease, n (%) | 9,031 (29.3) | 451 (27.8) | 0.033 | 374 (30.3) | 365 (29.6) | 0.016 |
| Aortic stenosis, % | 947 (3.1) | 42 (2.6) | 0.029 | 26 (2.1) | 29 (2.4) | 0.016 |
| Heart failure, n (%) | 14,398 (46.7) | 654 (40.3) | 0.13 | 561 (45.5) | 558 (45.3) | 0.005 |
| Renal failure, n (%) | 4,265 (13.8) | 108 (6.7) | 0.239 | 90 (7.3) | 84 (6.8) | 0.019 |
| Diabetes, n (%) | 9,591 (31.1) | 462 (28.5) | 0.058 | 381 (30.9) | 362 (29.4) | 0.034 |
| Anemia, n (%) | 1,635 (5.3) | 62 (3.8) | 0.071 | 71 (5.8) | 58 (4.7) | 0.047 |
| Risk of coagulation disorder, n (%) | 767 (2.5) | 27 (1.7) | 0.058 | 22 (1.8) | 22 (1.8) | <0.001 |
| Pre-operative heparin use, n (%) | 7,926 (25.7) | 337 (20.8) | 0.117 | 263 (21.3) | 272 (22.1) | 0.018 |
| Medication for anemia, n (%) | 3,087 (10.0) | 114 (7.0) | 0.107 | 116 (9.4) | 103 (8.4) | 0.037 |

IAD, intra-operative autologous blood donation; SD, standard deviation; SMD, standard mean difference. All covariates have a good balance when the SMD <0.1.

difference = -9.2%, p = .016). The secondary outcome of adverse events or post-operative ICU stay was not statistically different between the control and IAD groups (Tables 5 and 6). The total amounts of transfused RBCs, FFPs, and PLTs during hospitalization are summarized by surgical type in Fig 2. In the IAD group, the amounts of all transfusions in cardiac surgery and RBC and FFPs in aortic surgery were smaller than those in the control group.

## Discussion

This observational cohort study examined the association of IAD with the transfusion rate and amount for patients who had undergone cardiac and aortic surgeries using administrative data for the first time in Japan. We noted two main findings. First, IAD may decrease the transfusion rate of RBCs, FFPs, and PLTs in cardiac surgery patients and the transfusion rate of RBCs and FFPs in aortic surgery patients. Second, IAD can decrease the amounts of RBCs or FFPs without any worse outcomes or increased risk of surgical complications. Our findings suggested that IAD may be effective in the Japanese population, although postoperative adverse effects would not change.

**Table 2. Surgical information of the cardiac surgery patients before and after multilevel propensity score matching.**

|  | Before matching | | | After matching | | |
|---|---|---|---|---|---|---|
|  | **Control** | **IAD** | **SMD** | **Control** | **IAD** | **SMD** |
| n | 30,811 | 1,622 |  | 1,233 | 1,233 |  |
| Type of cardiac surgery, n (%) |  |  | 0.195 |  |  | 0.055 |
| Isolated CABG | 5,992 (19.4) | 231 (14.2) |  | 204 (16.5) | 180 (14.6) |  |
| Isolated valve repair | 20,288 (65.8) | 1,212 (74.7) |  | 891 (72.3) | 917 (74.4) |  |
| CABG + valve repair | 4,531 (14.7) | 179 (11.0) |  | 138 (11.2) | 136 (11.0) |  |
| Re-operation, n (%) | 1,161 (3.8) | 45 (2.8) | 0.056 | 38 (3.1) | 35 (2.8) | 0.014 |
| Total intra-operative CPB time (min), mean (SD) | 183.70 (88.02) | 186.30 (96.93) | 0.028 | 190.92 (94.19) | 186.99 (96.82) | 0.041 |

CABG, coronary artery bypass graft; CPB, cardiopulmonary bypass; IAD, intra-operative autologous blood donation; SD, standard deviation; SMD, standard mean difference. All covariates have a good balance when the SMD <0.1.

**Table 3. Baseline characteristics and comorbidities of the aortic surgery patients before and after multilevel propensity score matching.**

| | Before matching | | | After matching | | |
|---|---|---|---|---|---|---|
| | **Control** | **IAD** | **SMD** | **Control** | **IAD** | **SMD** |
| n | 3,983 | 284 | | 197 | 197 | |
| Sex (male), n (%) | 2,860 (71.8) | 243 (85.6) | 0.341 | 162 (82.2) | 167 (84.8) | 0.068 |
| Age (y), mean (SD) | 71.14 (9.38) | 70.10 (8.66) | 0.115 | 69.80 (10.65) | 70.73 (8.81) | 0.096 |
| Body weight (kg), mean (SD) | 63.27 (11.83) | 66.61 (12.08) | 0.279 | 65.17 (10.85) | 65.98 (11.66) | 0.072 |
| Ischemic heart disease, n (%) | 1,129 (28.3) | 78 (27.5) | 0.02 | 56 (28.4) | 56 (28.4) | <0.001 |
| Aortic stenosis (%) | 30 (0.8) | 1 (0.4) | 0.054 | 0 (0.0) | 0 (0.0) | <0.001 |
| Heart failure, n (%) | 964 (24.2) | 62 (21.8) | 0.056 | 41 (20.8) | 48 (24.4) | 0.085 |
| Renal failure, n (%) | 259 (6.5) | 13 (4.6) | 0.084 | 12 (6.1) | 10 (5.1) | 0.044 |
| Diabetes, n (%) | 856 (21.5) | 61 (21.5) | <0.001 | 45 (22.8) | 47 (23.9) | 0.024 |
| Anemia, n (%) | 178 (4.5) | 8 (2.8) | 0.088 | 10 (5.1) | 8 (4.1) | 0.049 |
| Risk of coagulation disorder, n (%) | 156 (3.9) | 7 (2.5) | 0.083 | 3 (1.5) | 3 (1.5) | <0.001 |
| Pre-operative heparin use, n (%) | 580 (14.6) | 46 (16.2) | 0.045 | 30 (15.2) | 34 (17.3) | 0.055 |
| Medication for anemia, n (%) | 140 (3.5) | 9 (3.2) | 0.019 | 9 (4.6) | 9 (4.6) | <0.001 |

IAD, intra-operative autologous blood donation; SD, standard deviation; SMD, standard mean difference. All covariates have a good balance when the SMD <0.1.

Our main result contradicts the findings by previous studies that reported negative results for IAD in CVS [20, 21]. One of the reasons is that we stratified the patients into cardiac surgery and aortic surgery, and analyzed them separately, different from the previous studies [20, 21]. From a clinical perspective, aortic surgery with longer CPB time and lower CPB temperature than cardiac surgery should not be analyzed in the same category. Additionally, our study showed well-balanced CPB time for both surgery types, based on multilevel propensity score matching. CPB duration helps predict peri-operative bleeding or transfusion [19]. However, our results are consistent with some recent studies [13, 14, 22]. Since the eligible patients' body weight and BSA were high, a large IAD volume was recommended to reduce transfusions [13, 14, 22]. Our target population is the Japanese patients, who are physically smaller than individuals in Western countries; hence, it is difficult to perform high-volume IAD with the same strategy as reported in the previous studies [14, 15]. Although the average amount of IAD was smaller in our eligible patients, the average body weight was also smaller, and we believed that we obtained positive results.

**Table 4. Surgical information of the aortic surgery patients before and after multilevel propensity score matching.**

| | Before matching | | | After matching | | |
|---|---|---|---|---|---|---|
| | **Control** | **IAD** | **SMD** | **Control** | **IAD** | **SMD** |
| n | 3,983 | 284 | | 197 | 197 | |
| Type of aortic surgery, n (%) | | | 0.192 | | | 0.032 |
| Isolated ascending aorta repair | 513 (12.1) | 38 (10.9) | | 22 (11.2) | 23 (11.7) | |
| Isolated arch repair | 1937 (45.9) | 131 (37.6) | | 78 (39.6) | 75 (38.1) | |
| Total arch repair | 1773 (42.0) | 179 (51.4) | | 97 (49.2) | 99 (50.3) | |
| Re-operation, n (%) | 3 (0.1) | 0 (0.0) | 0.038 | 0 (0.0) | 0 (0.0) | <0.001 |
| Intra-operative CPB time (≥32˚C) (min), mean (SD) | 45.87 (78.26) | 35.97 (69.56) | 0.134 | 21.81 (52.59) | 26.67 (56.90) | 0.089 |
| Intra-operative CPB time (<32˚C) (min), mean (SD) | 173.25 (89.91) | 180.40 (75.45) | 0.086 | 179.45 (77.27) | 182.37 (75.13) | 0.038 |

IAD, intra-operative autologous blood donation; CPB, cardiopulmonary bypass; SD, standard deviation; SMD, standard mean difference. All covariates have a good balance when the SMD <0.1.

**Table 5. Outcomes of the cardiac surgery patients before and after multilevel propensity score matching.**

| Cardiac surgery | Before matching | | After matching | | | |
| --- | --- | --- | --- | --- | --- | --- |
| | Control | IAD | Control | IAD | R (D) | p-value |
| n | 30,811 | 1,622 | 1,233 | 1,233 | | |
| RBC transfusion rate, % | 76.7 | 40.6 | 60.6 | 38.4 | -22.2 | <0.001 |
| FFP transfusion rate, % | 71.8 | 37.3 | 53.1 | 34.4 | -18.7 | <0.001 |
| PLT transfusion rate, % | 42.9 | 19.1 | 27.7 | 18.6 | -9.1 | <0.001 |
| Fibrinogen use, % | 0.5 | 0.4 | 0.6 | 0.3 | -0.3 | 0.546 |
| Re-operation for bleeding, % | 1.8 | 1.3 | 1.2 | 1.4 | 0.2 | 0.86 |
| Post-operative AKI, % | 3.1 | 2 | 3.7 | 2.4 | -1.3 | 0.081 |
| Post-operative respiratory failure, % | 3.1 | 1.2 | 1.4 | 1.6 | 0.2 | 0.735 |
| Post-operative infection, % | 7.5 | 5.9 | 5.4 | 5.5 | 0.1 | 1 |
| Post-operative ICU stay (days), mean (SD) | 3.11(2.73) | 2.75(2.09) | 2.78(2.48) | 2.61(2.12) | -0.17 | 0.07 |

AKI, acute kidney injury; FFP, fresh frozen plasma; IAD, intra-operative autologous blood donation; ICU, intensive care unit; P LT, platelet; RBC, red blood cell; SD, standard deviation. Effect sizes are presented as the risk difference. The p-values were estimated by using paired $t$ test for continuous variables and the McNemar test for categorical variables.

Unlike IAD, intra-operative cell salvage allows intra-operative blood to be collected and concentrated RBC components to be reinfused. However, intra-operative hemorrhage can cause loss of coagulation-related factors and platelet aggregation function [23]. IAD could contribute to the reduction of allogeneic blood component transfusions for RBC mass or hemostatic elements of whole blood. In CVS patients, autologous blood collected before CPB initiation is generally stored at room temperature and used intra-operatively. Therefore, IAD can provide coagulation-related factors (PLTs, fibrinogen, plasmin–antiplasmin complex, and antithrombin) within the normal range without deactivation, which is a substantial advantage [24].

In this study, we have two major strengths. First, the DPC database, which covers most acute care hospitals in Japan [25], enabled us to enroll many patients in this study. Therefore, our findings on blood transfusion for scheduled CVS are likely to reflect the current situation

**Table 6. Outcomes of the aortic surgery patients before and after multilevel propensity score matching.**

| Aortic surgery | Before matching | | After matching | | | |
| --- | --- | --- | --- | --- | --- | --- |
| | Control | IAD | Control | IAD | R (D) | p-value |
| n | 3,983 | 284 | 197 | 197 | | |
| RBC transfusion rate, % | 94.8 | 83.1 | 91.4 | 83.8 | -7.6 | 0.037 |
| FFP transfusion rate, % | 96 | 81.7 | 91.9 | 82.7 | -9.2 | 0.016 |
| PLT transfusion rate, % | 88.8 | 78.2 | 76.6 | 76.1 | -0.5 | 1 |
| Fibrinogen use, % | 1.2 | 1.4 | 0.5 | 2 | -0.5 | 1 |
| Re-operation for bleeding, % | 2.4 | 1.4 | 2.5 | 2 | -0.5 | 1 |
| Post-operative AKI, % | 3 | 5.6 | 8.1 | 5.1 | -3 | 0.307 |
| Post-operative respiratory failure, % | 3.5 | 4.2 | 4.6 | 2.5 | -2.1 | 0.386 |
| Post-operative infection, % | 10.3 | 9.5 | 11.7 | 8.6 | -3.1 | 0.417 |
| Post-operative ICU stay (days), mean (SD) | 3.60(3.16) | 3.08(2.45) | 3.21(3.18) | 3.24(2.55) | -0.06 | 0.834 |

AKI, acute kidney injury; FFP, fresh frozen plasma; IAD, intra-operative autologous blood donation; ICU, intensive care unit; PLT, platelet; RBC, red blood cell; SD, standard deviation. Effect sizes are presented as the risk difference. The p-values were estimated by using paired $t$ test for continuous variables and the McNemar test for categorical variables.

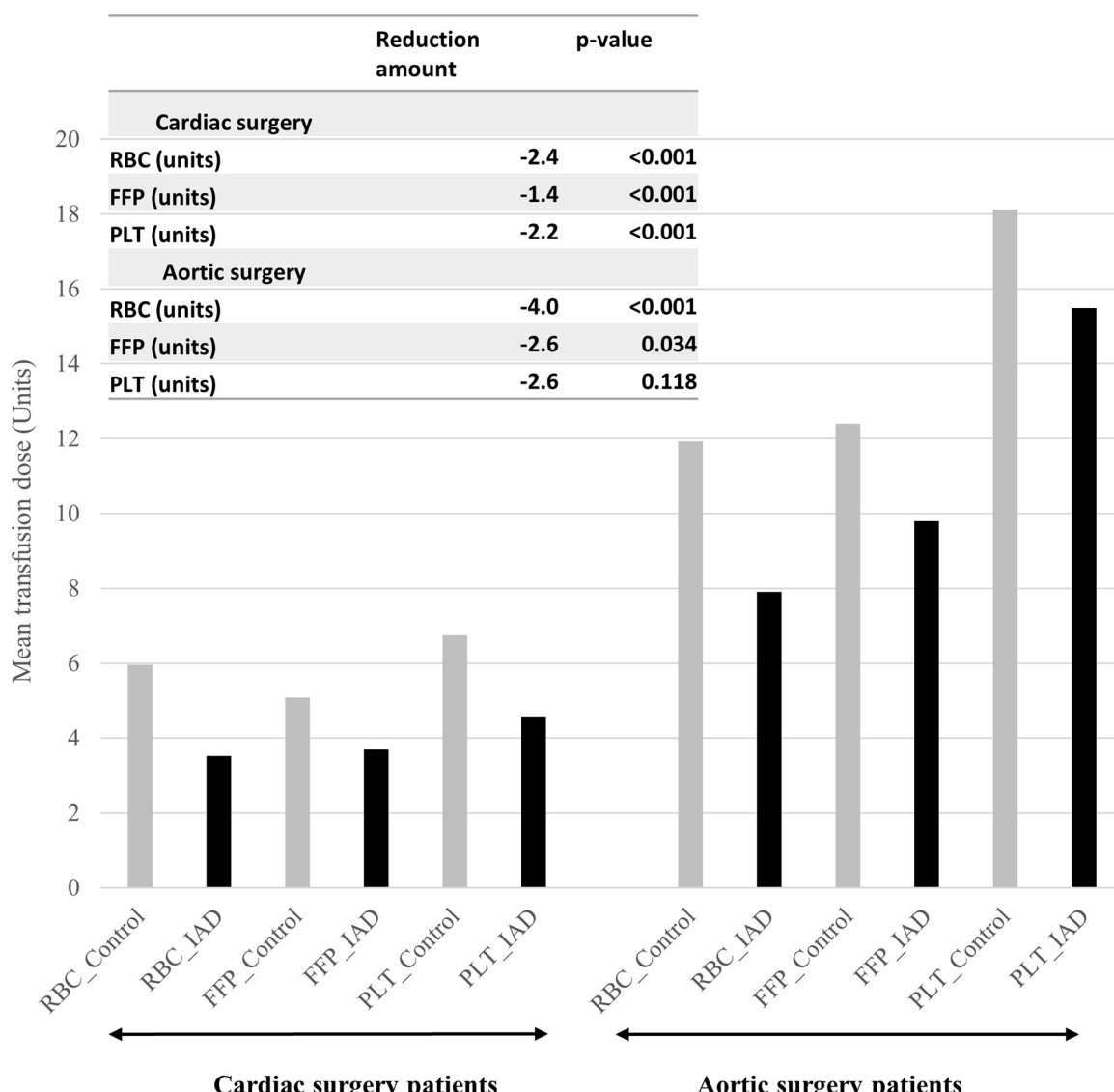

| | Reduction amount | p-value |
|---|---|---|
| **Cardiac surgery** | | |
| **RBC (units)** | **-2.4** | **<0.001** |
| **FFP (units)** | **-1.4** | **<0.001** |
| **PLT (units)** | **-2.2** | **<0.001** |
| **Aortic surgery** | | |
| **RBC (units)** | **-4.0** | **<0.001** |
| **FFP (units)** | **-2.6** | **0.034** |
| **PLT (units)** | **-2.6** | **0.118** |

**Fig 2. Total units of transfused blood components during hospitalization, based on surgical type.** FFP, fresh frozen plasma; IAD, intra-operative autologous blood donation; PLT, platelet; RBC, red blood cell.

in Japan. Our results are based on better-balanced populations than previous studies that were limited by selection bias and insufficient propensity score matching (SMD > 0.1) [14, 15]. Second, to eliminate confounding bias, we calculated the propensity score using patient-level variables, based on the protocol of previous studies [26] as the first layer and hospital ID as the second layer, resulting in a multilevel propensity score analysis. To date, few studies have conducted propensity score analyses based on a clustered data structure [26]. However, to analyze clustered data, especially when the intention to treat strongly depends on hospitals [26], multilevel regression models should be considered. By using this multilevel propensity score method, we managed to adjust the hospital differences of intention to perform IAD.

However, we had several limitations. First, we cannot rule out the effect of unmeasured confounding bias, including the amount of blood loss and more detailed CPB information, which may skew our results; nevertheless, we used a multilevel propensity score analysis with

good balance and examined significance using the McNemar test. Second, because of the database's characteristics, we could not consider the baseline hematocrit level or details of CPB. The IAD may be rarely used for patients with severe anemia or a small BSA [15]. Instead, we substituted the diagnosis of anemia and coagulation disorders as comorbidities on the operative day and the use of pre-operative medication for anemia. The intention of IAD can differ based on body weight or BSA. However, the large sample size of our study and well-balanced propensity score and variables strengthen our findings. Compared to the previous studies, our study showed a higher rate of allogenic transfusion and a lower rate of IAD use. IAD was approved in 2016 by the Ministry of Health, Labour and Welfare of Japan, and this could explain the low number of IAD cases in our study. However, a prolonged COVID-19 pandemic could lead to chronic insufficiency of transfusion blood supply, and more widespread use of IAD is expected with the accumulation of evidence from other regions in Japan. We believe that our report will guide the increased use of IAD in Japan.

Concluding, this study demonstrated that IAD could reduce the transfusion rate or the amount of allogeneic blood transfusions, especially RBC and FFP, in patients who underwent CVS. Despite the limitations, the present findings suggest that IAD is an effective blood transfusion strategy in Japan, especially during blood shortage owing to COVID-19. Future research should examine the effect of the dose-response efficiency of IAD in physically small people, such as the Japanese. The postoperative adverse events, which are the most important outcomes cited as major reasons for avoiding autologous blood product administration, should also be examined.

## Acknowledgments

We thank all staff members and all participating hospitals.

## Author Contributions

**Conceptualization:** Takuya Okuno, Yuichi Imanaka.

**Data curation:** Takuya Okuno, Susumu Kunisawa, Kiyohide Fushimi, Yuichi Imanaka.

**Formal analysis:** Takuya Okuno.

**Funding acquisition:** Yuichi Imanaka.

**Methodology:** Takuya Okuno.

**Project administration:** Takuya Okuno.

**Software:** Takuya Okuno, Susumu Kunisawa.

**Supervision:** Takuya Okuno.

**Validation:** Takuya Okuno.

**Visualization:** Takuya Okuno.

**Writing – original draft:** Takuya Okuno.

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
