## [Decision Letter · Decision Letter 0]

23 Dec 2020

PONE-D-20-38371

Intra-operative autologous blood donation for cardiovascular surgeries in Japan: a retrospective cohort study

PLOS ONE

Dear Dr. Imanaka,

Thank you for submitting your manuscript to PLOS ONE. After careful consideration, we feel that it has merit but does not fully meet PLOS ONE’s publication criteria as it currently stands. Therefore, we invite you to submit a revised version of the manuscript that addresses the points raised during the review process.

We look forward to receiving your revised manuscript.

Kind regards,

Arman Kilic

Academic Editor

PLOS ONE

Additional Editor Comments:

This is an interesting and important analysis demonstrating potential benefit of intra-operative autologous blood donation for cardiac surgery. Please address the reviewers suggestions and comments.

4. Thank you for providing the date range when patients has surgery. Please also include the date(s) on which you accessed the databases or records to obtain the retrospective data used in your study.

Reviewers' comments:

Reviewer's Responses to Questions

**Comments to the Author**

1. Is the manuscript technically sound, and do the data support the conclusions?

Reviewer #1: Yes

Reviewer #2: Yes

2. Has the statistical analysis been performed appropriately and rigorously? 

Reviewer #1: Yes

Reviewer #2: Yes

3. Have the authors made all data underlying the findings in their manuscript fully available?

Reviewer #1: No

Reviewer #2: Yes

4. Is the manuscript presented in an intelligible fashion and written in standard English?

Reviewer #1: No

Reviewer #2: No

5. Review Comments to the Author

Reviewer #1: The authors are congratulated for submitting excellent work regarding the effects of intraoperative autologous blood donation on the rate and volume of blood transfusion, using an administrative database covering the major cardiac surgery centers in Japan. The authors have appraised much of the existing literature and wished to examine whether the practice of IAD is beneficial in the Japanese population, who tend to be smaller and less obese than previously studied populations. The chief finding is that use of IAD in Japan is associated with fewer transfusions of RBC, FFP, and Platelets in non-aortic cardiac surgery, and with fewer transfusions of RBC's and FFP in aortic surgery, with no increase in the examined complications.

The analysis is retrospective and relies on multi-layered propensity matching based on patient characteristics and the treating hospital. The statistical analysis appears robust, and the conclusions and analysis are well done. I think the authors correctly conclude from this data that IAD can lessen the amount of transfusion needed in cardiac surgery without increasing major complications. The aortic surgery subgroup analysis is underpowered to make any conclusion beyond the reduction of FFP usage.

The work is original and generally well done. The major area in need of revision is the discussion section.

Edits by line number:

52-53: please clarify syntax. Presumably minimizing transfusion is also important for surgeons and patients, too.

58: syntax needs to be clarified

116-117: please revise the sentence for clarity -- presuming that heparin and antiplatelet agents were stopped in the usual fashion

118-120: Standard accepted definitions for hypothermia should be followed. Deep is <24C, Moderate 24-32C, mild >32C. (Leshnower et al, 2015 Annals Thoracic Surgery)

150: please edit "arterial" to read "aortic"

166-167: please revise for clarity.

Discussion section is not page numbered. There are multiple areas in the discussion that need copy editing for clarity. When you state that IAD was only approved in 2016 -- approved by whom? The discussion touches on perceived shortcomings of other peer-reviewed work. Please revise to focus instead on what the present study demonstrates.

Reviewer #2: 1. Please ask English native speaker to review and check the manuscript.

2. Title of the front page has to be same as the title of the manuscript. Do not exclude “in Japan” in the title.

3. What is “control group”? I guess it is the patient group without IAD, but you have to mention it.

4. Why did you exclude the cases with CPB time less than 40 minutes? I don’t know what you mean by “indicating possible intra-operative challenges (<40min)”? Why not 30 minutes, why 40 minutes?

5. Did you include the emergency cases? These tend to have more transfusion in general.

6. Did you include off-pump CABG? These tend to have less transfusion in general.

7. Why do you think “Re-operative for bleeding” of the IAD group is higher than control group in aortic surgery patients? IAD group receive less transfusion, but higher “re-operation for bleeding”? I know it is not statistically significant, but please let me know your opinion.

8. Some control group patients received “PAD”. Do you really think these patients can be in the control group? It is still the autologous blood donation, so PAD may contribute to decrease the transfusion rate, so I think these patients should be considered separately, or excluded.

9. What do you think is the “unmeasured confounding bias”? Please mention the examples.

6. PLOS authors have the option to publish the peer review history of their article (what does this mean?). If published, this will include your full peer review and any attached files.

Reviewer #1: No

Reviewer #2: No

---

## [Author Response · Author response to Decision Letter 0]

16 Jan 2021

We thank you and the reviewers for your thoughtful suggestions and insights. The manuscript has benefited from these insightful suggestions. I look forward to working with you and the reviewers to move this manuscript closer to publication in PLOS ONE. The necessary changes have been made in accordance with the reviewers’ suggestions. The responses to all comments have been prepared and attached in a separate file, "Response to Reviewers".

---

## [Editor Report · Decision Letter 1]

4 Feb 2021

Intra-operative autologous blood donation for cardiovascular surgeries in Japan: a retrospective cohort study

PONE-D-20-38371R1

Dear Dr. Imanaka,

We’re pleased to inform you that your manuscript has been judged scientifically suitable for publication and will be formally accepted for publication once it meets all outstanding technical requirements.

Kind regards,

Arman Kilic

Academic Editor

PLOS ONE

Additional Editor Comments (optional):

The authors have responded satisfactorily to the comments. This represents an important contribution.
---

## [Editor Report · Acceptance letter]

10 Feb 2021

PONE-D-20-38371R1 

Intra-operative autologous blood donation for cardiovascular surgeries in Japan: a retrospective cohort study 

Dear Dr. Imanaka:

I'm pleased to inform you that your manuscript has been deemed suitable for publication in PLOS ONE. Congratulations! Your manuscript is now with our production department. 

Kind regards, 

on behalf of

Dr. Arman Kilic 

Academic Editor

PLOS ONE